# Recombinant *Toxoplasma gondii* Ribosomal Protein P2 Modulates the Functions of Murine Macrophages In Vitro and Provides Immunity against Acute Toxoplasmosis In Vivo

**DOI:** 10.3390/vaccines9040357

**Published:** 2021-04-07

**Authors:** Zhengqing Yu, Yujia Lu, Zhaoyi Liu, Muhammad Tahir Aleem, Junlong Liu, Jianxun Luo, Ruofeng Yan, Lixin Xu, Xiaokai Song, Xiangrui Li

**Affiliations:** 1MOE Joint International Research Laboratory of Animal Health and Food Safety, College of Veterinary Medicine, Nanjing Agricultural University, Nanjing 210000, China; 2018207044@njau.edu.cn (Z.Y.); 11118310@njau.edu.cn (Y.L.); 17118306@njau.edu.cn (Z.L.); 2018207076@njau.edu.cn (M.T.A.); yanruofeng@njau.edu.cn (R.Y.); xulixin@njau.edu.cn (L.X.); songxiaokai@njau.edu.cn (X.S.); 2State Key Laboratory of Veterinary Etiological Biology, Key Laboratory of Veterinary Parasitology of Gansu Province, Lanzhou Veterinary Research Institute, Chinese Academy of Agricultural Sciences, Lanzhou 730046, China; liujunlong@caas.cn (J.L.); luojianxun@caas.cn (J.L.)

**Keywords:** *Toxoplasma gondii*, ribosomal protein P2, murine macrophage, immune protection

## Abstract

Almost every warm-blooded animal can be an intermediate host for *Toxoplasma gondii* (*T. gondii*); there is still no efficient vaccine and medicine available for *T. gondii* infections. Detected on the surface of free tachyzoites of *T. gondii*, *T. gondii* ribosomal protein P2 (TgRPP2) has been identified as a target for protection against toxoplasmosis. In the present study, TgRPP2 was firstly expressed in a prokaryotic expression system, and the purified recombinant TgRPP2 (rTgRPP2) was characterized by its modulation effects on murine macrophages. Then, the purified rTgRPP2 was injected into mice to evaluate the immune protection of rTgRPP2. The results indicated that rTgRPP2 could bind to murine Ana-1 cells and showed good reactogenicity. After incubation with purified rTgRPP2, the proliferation, apoptosis, phagocytosis, nitric oxide (NO) production, and cytokines secreted by murine macrophages were modulated. Furthermore, the in vivo experiments indicated that animals immunized with rTgRPP2 could generate a significantly high level of antibodies, cytokines, and major histocompatibility complex (MHC) molecules, leading to a prolonged survival time. All of the results indicated that murine macrophages could be regulated by rTgRPP2 and are essential for the maintenance of tissue homeostasis. Immunization with rTgRPP2 triggered significant protection, with prolonged survival time in a mice model of acute toxoplasmosis. Our results lend credibility to the idea that rTgRPP2 could be a potential target for drug design and vaccine development.

## 1. Introduction

*Toxoplasma gondii* (*T. gondii*) is an obligate intracellular parasite belonging to the phylum Apicomplexa. It can infect a wide range of warm-blooded animals, exhibiting zoonotic potential and being responsible for human toxoplasmosis [1,2]. According to a conservative estimation, more than 1 billion people have been infected around the world [2,3]. Opportunistic parasites are usually symbiotic organisms that become pathogenic in specific conditions, mainly in immunocompromised hosts (e.g., AIDS patients). *T. gondii* is not an opportunistic pathogen; it is a true pathogen that can cause disease itself, even without favorable conditions [4]. Normally, immunocompetent patients do not display symptoms. However, these parasites can cause severe disease, even leading to death in immunocompromised individuals [5]. According to previous reports, *T. gondii* can be spread by the ingestion of unsterilized meat and contaminated fruits, vegetables, and drinking water [6,7,8]. Moreover, the vertical transmission of *T. gondii* can cause irreversible damage to a fetus, miscarriage, and stillbirth; additionally, congenital deformities can occur [9,10,11,12]. Currently, there are no effective vaccines and treatments for *T. gondii*. Based on the *T. gondii* S48 strain, a live attenuated vaccine (Ovilis Toxovax®, Intervet, Angers, France) has been approved for sheep and goats [13]. However, the toxicity, mechanism, and stability of Ovilis Toxovax® still need to be improved. Pyrimethamine (PYR) and sulfadiazine (SDZ) have been approved for the treatment of toxoplasmosis. However, these two drugs can only depress Toxoplasma folate synthesis [14] and do not affect the bradyzoites, suggesting that they cannot eliminate chronic infection [15]. Moreover, the side effects, such as kidney disorders, immunosuppression, and teratogenicity in the fetus, are also obvious [16,17]. Conclusively, the development of an effective prevention and treatment therapy against toxoplasmosis is an urgent and important requirement.

Cell immunity reaction is an efficient way for hosts to resist *T. gondii* [18]. The activated macrophages have been divided into two main phenotypes: classically activated macrophages (M1) and alternatively activated macrophages (M2) [19,20]. An increasing number of reports have suggested that macrophages in different phenotypes present different functions in the inflammatory response and tissue repair [21,22]. Many proinflammatory agents are generated during early and acute *T. gondii* invasion, and, largely, cytokines are released. Cytokine IFN-γ can efficiently activate M1 macrophages through the NF-κB signaling pathway and promote M1 macrophages to secrete proinflammatory cytokines (IL-1β, TNF-α, IL-6, and IL-23), inducible nitric oxide synthase (iNOS), and nitric oxide (NO), even enhancing the ability of phagocytes to resist *T. gondii* [23,24]. Interestingly, although macrophages play an important role in resisting the replications of *T. gondii* [25], *T. gondii* has developed multistrategies to limit the antimicrobial activities of macrophages [26,27]. Thus, developing an appropriate and efficient target may prove a highly efficacious strategy in the development of *T. gondii* vaccines.

As is known, a complete ribosome is composed of a large subunit (LSU) and a small subunit (SSU). In eukaryotes, P-proteins comprise the large subunit, with the P0 protein in the center, finally forming the stalk P0-(P1-P2)_2_ complex with the ribosomal proteins P1 and P2 [28]. Ribosomes and mitoribosomes are considered to have a fundamental role in parasite biology. According to previous reports, some genes encoding mitochondrial protein are crucial for the survival of *T. gondii* [29]. Based on this fact, mitochondria and mitoribosomes have been the focus of drug and vaccine discovery research [30,31,32], and there is evidence that ribosomal proteins are promising targets for anti-*T. gondii* infections [33]. A recent study showed that the ribosomal P2 protein and the P protein pentamer complex were immunogenic in *Plasmodium falciparum* [34], and the P2 protein from *P. falciparum* was highly homologous to the *T. gondii* RPP2 protein (TgRPP2). Unlike the ribosomal P2 protein of *P. falciparum* [35,36], the TgRPP2 protein has been demonstrated to exist on the surface of *T. gondii* tachyzoites using the immunohistochemical method; however, the TgRPP2 protein was not observed on the surface of human foreskin fibroblast (HFF) cells during *T. gondii* infections in vitro [33]. This finding strengthens the hypothesis that TgRPP2 could be involved in *T. gondii* invasion. Therefore, it could be a potential target for effective prevention and treatment therapy against *T. gondii*. Although signal and transmembrane domains have not been reported for the TgRPP2 protein [35], there are still some reports suggesting that TgRPP2 might be involved as an invasion ligand, which could help *T. gondii* adhere to host cells [37].

Using a prokaryotic expression system to further explore how recombinant TgRPP2 (rTgRPP2, Uniprot ID: A0A125YFT4) modulates host immunity, we employed an unbiased approach to investigate the potential effects of rTgRPP2 on murine macrophages in vitro and immune protection against acute toxoplasmosis in vivo. The in vitro results showed that rTgRPP2 could enhance proliferation, expedite apoptosis, stimulate phagocytosis, promote NO secretion, and modulate cytokine secretion. The in vivo results revealed that rTgRPP2 could provide immune protection and prolong survival time against a lethal challenge with the virulent RH strain of *T. gondii*. All these results indicate that *Toxoplasma* ribosomal protein P2 could be a good candidate for *T. gondii* prevention and treatment.

## 2. Materials and Methods

### 2.1. Animals, Parasites, and Cultured Cells

Sprague Dawley (SD) rats (male), weighing 200–220 g, were bought from the Center of Comparative Medicine, Yangzhou University, China, and grown in a specific pathogen-free environment. *T. gondii* type I (RH strain) was kept in the Ministry of Education (MOE) Joint International Research Laboratory of Animal Health and Food Safety, College of Veterinary Medicine, Nanjing Agricultural University, Nanjing, Jiangsu, PR China. To maintain these parasites, previous methods described by Zhao were conducted [38]. The mouse cell line of Ana-1 macrophages, which was received from the Institute of Cell Biology, Chinese Academy of Sciences (Shanghai, China), was cultivated in Roswell Park Memorial Institute 1640 (RPMI 1640) culture medium supplemented with heat deactivated 10% FBS (fetal bovine serum, Gibco, Grand Island, NY, USA), with 1% penicillin/streptomycin at 37 °C in a CO_2_ (5%) standard atmosphere.

### 2.2. Cloning and Molecular Characterization of TgRPP2

In accordance with the manufacturer’s manual, Trizol reactant (Invitrogen, Shanghai, China) was applied to extract the whole RNA from tachyzoites of *T. gondii*. After this, cDNA was manufactured by utilizing the cDNA Kit (Takara Biotechnology, Dalian, China) and retained at −80 °C. Primers were prepared and designed based on the coding sequence (CDS) of the TgRPP2 gene (GeneBank: XM_002364187). Restraint endonuclease *Eco* RI and *Hind* III (Takara Biotechnology, Dalian, China) were placed in sense and antisense primers successively, and homologous sequences were also included, following the manufacturer’s guidance (Vazyme Biotech, Nanjing, China). The sense and antisense primers, 5′-GCTGATATCGGATCC GAATTC ATGGCAATGAAATACTTCGCTG-3′ and 5′-CTCGAGTGCGGCCGC AAGCTT TTAGTCGAAGAGCGAGAAGCCC-3′, targeting the complete 342 bp TgRPP2 open reading frame (ORF), were obtained from Tsingke Biological Technology (Nanjing, China). The amplification reaction mixture consisted of 1.25 U *EX Taq* (Takara Biotechnology, Dalian, China), 5 µL 10 × *Ex Taq* buffer, 4 µL dNTP mixtures (Mg^2+^ plus), 20 pmol of both primers, and a 2-ng cDNA model in a final volume of 50 µL. The PCR condition was 95 °C for 5 min, followed by 35 cycles of 95 °C for 30 s, 62 °C for 30 s, and 72 °C for 1 min, and the last extension step at 72 °C for 5 min. Then, the PCR products were electrophoresed in a 1.0% tris-borate-EDTA (TBE) buffer and stained with Goldview (Solarbio, Beijing, China). The amplicons were visualized under UV light and purified by the E.Z.N.A. Gel-Extraction Kit (Omega Biotech, Norcross, GA, USA), according to the manufacturer’s directions. Finally, the amplicons were subcloned into a linear pET32a vector (Takara Biotechnology, Dalian, China) that had been digested by restriction endonucleases *Eco* RI and *Hind* III using the One Step Cloning Kit (Vazyme Biotech, Nanjing, China). After that, recombinant plasmid pET32a/TgRPP2 was transformed into DH5α (Invitrogen Biotechnology, Shanghai, China) and cultivated in Luria Bertani culture medium (LB) with a dosage of 100 μg/mL ampicillin. The recombinant plasmid pET32a/TgRPP2 was then determined by double enzyme digest and sequenced on an ABI PRISM™ 3730 XL DNA analyzer (Applied Biosystems, Carlsbad, CA, USA) by Tsingke Biological Technology (Nanjing, China). Sequence analysis was also conducted using the online Blast system (https://blast.ncbi.nlm.nih.gov/Blast.cgi, accessed on 6 April 2021).

### 2.3. Expression and Purification of Recombinant TgRPP2 and the pET32a Vector Protein

The verified recombinant plasmid was isolated under the guidance of the Plasmid-Mini Kit (Omega Biotech, Norcross, GA, USA) and transformed into *Escherichia coli* BL21 (DE3) (Invitrogen Biotechnology, Shanghai, China). To obtain the pET32a vector protein, the circular pET32a obtained from the dealers was also transformed into *E. coli* BL21 (DE3). Recombinant TgRPP2 and the pET32a vector protein were expressed under the induction of 1 mM isopropyl-*β*-d-thiogalactopyranoside (IPTG; Sigma-Aldrich, Shanghai, China) and purified with a chelating nickel column (Ni-NTA, GE Healthcare, Piscataway, NJ, USA) in accordance with the instructions. The recombinant protein was verified by 12% sodium salt polyacrylamide gel electrophoresis (SDS-PAGE) and stained with Coomassie blue. Endotoxins in the recombinant protein were removed according to the directions of the ToxinEraser™ Endotoxin Removal Kit (GeneScript, Piscataway, NJ, USA). Before further analysis, the ToxinSensor™ Chromogenic LAL Endotoxin Assay Kit (GeneScript, Piscataway, NJ, USA) was also used to detect the level of endotoxins in the recombinant proteins. The concentration of recombinant protein was also determined by using the Pierce™ BCA Protein Assay Kit (Thermo Scientific, Waltham, MA, USA). Briefly, 25 μL of each standard or recombinant protein was added into a microplate well; then, 200 μL of working reagent (50 part “Reagent A” and 1 part “Regent B”) was added to each well and mixed in the microplate thoroughly. After incubation at 37 °C for 30 min in the dark, protein concentration was measured by measuring the absorbance at 562 nm using a plate reader (Thermo Scientific, Waltham, MA, USA).

### 2.4. Total Soluble Protein of T. gondii Tachyzoites

The purified *T. gondii* tachyzoites were collected in 1 mL pH 7.4 phosphate buffer solution. After three freeze–thaw steps, the tachyzoites were then crushed by sonication at a continuous mode for 2 s at 4 s intervals (8 min in total) under high power (60 W) on ice. A RIPA solution (Thermo Scientific, Waltham, MA, USA) and a proteinase inhibitor (Thermo Scientific, Waltham, MA, USA) were then added for the complete lysis of proteins. The solution, consisting of the total soluble protein of *T. gondii* tachyzoites, was collected by high-speed centrifugation at 12,000 r/min for 30 min at 4 °C. Then, the supernatant was determined using the bicinchoninic acid method (BCA) mentioned above.

### 2.5. Western Blot Analysis of the pET32a Vector Protein, Recombinant, and Native TgRPP2

To obtain serum from SD rats immunized by rTgRPP2, 200 µg of purified rTgRPP2 was emulsified with an equal volume of Freund’s complete adjuvant (Sigma-Aldrich, Saint Louis, USA) to produce an emulsion vaccine, and the SD rats were then immunized with the emulsion vaccine as the first injection. Applying two-week intervals, the emulsion mixture, composed of 200 µg purified rTgRPP2 and an equal volume of Freund’s incomplete adjuvant (Sigma-Aldrich, Saint Louis, MO, USA), was injected into rats as the second to fourth injections. All immunized emulsions were subcutaneously delivered into the back skin of rats. Whole blood was collected from the orbit one week after the last immunization, and the serum was then separated.

The *T. gondii* infection serum was also harvested from SD rats artificially challenged with 5 × 10^5^
*T. gondii* tachyzoites by intra-abdominal infection three weeks before. Blank serum was also harvested from the SD rats as the negative control. All the serums were kept at −20 °C until use, and all of the SD rats were kept in the same environment. *T. gondii* lysates were also prepared, as previously described [39].

The purified recombinant TgRPP2 protein, pET32a vector protein, and *T. gondii* lysates were first separated by 12% SDS-PAGE gel and then transferred to a polyvinylidene difluoride (PVDF) membrane (Immobilon-PSQ, Millipore, Billerica, MA, USA). Initially, a blocking buffer containing 5% (*w/v*) skimmed milk powder dissolved in TBST (TBS containing 0.5% Tween 20) was used to block nonspecific binding sites at 37 °C for 2 h. Then, after being bathed in TBST three times, the membrane was incubated in primary antibodies (1:100 dilution, different sera collected from rats) at 37 °C for 2 h. Diluted at 1:5000, according to the guidelines of instruction, the membrane was then incubated with HRP-conjugated goat anti-rat IgG (eBioscience, San Diego, CA, USA) in TBST at 37 °C for 1 h, after being washed three times in TBST. To detect pET32a vector protein, purified recombinant TgRPP2 protein and pET32a vector protein were also analyzed by the mouse anti-His-tagged antibody (Proteintech Group, Rosemont, IL, USA) and HRP-conjugated goat anti-mouse IgG (Biosharp life sciences, Hefei, China) as the primary and secondary antibodies. Finally, the membrane was stained with freshly prepared 3,3’-diaminobenzidine (DAB, Sigma-Aldrich, Saint-Louis, MO, USA) to identify the bound antibodies.

### 2.6. Confirmation of rTgRPP2 Binding with Murine Macrophages

In order to verify the combination ability of recombinant TgRPP2 with murine macrophages, 1 × 10^6^ Ana-1 cells were collected in a 12-well plate (Costar, Cambridge, MA, USA) and incubated with phosphate-buffered solution (PBS), pET32a vector protein (20 μg/mL) and rTgRPP2 (20 μg/mL) at 37 °C for 1 h. After being washed three times in PBS to remove uncombined protein, the cells were then sequentially exposed to the sera from rats immunized by rTgRPP2 (1:100 dilutions) and mouse anti-rat IgG tagged with FITC (1:500 dilution; eBioscience, San Diego, CA, USA); each incubation step was performed at 37 °C for 15 min. After having been bathed in PBS three times to remove the uncombined antibodies, the murine macrophages were then analyzed by flow cytometry (Beckman Coulter, Brea, CA, USA). Forward scatter (FSC) and side scatter (SSC) were taken to gate the macrophage subsets [40]. Based on the circled cells, a linear gate was then determined according to the blank samples (PBS group). The independent experiments were repeated three times, and each sample was tested once. The percentage was then calculated.

### 2.7. Cell Proliferation Detection Assay

The Ana-1 cells were diluted to 5 × 10^5^ cells per well in a 96-well plate. After having been incubated with PBS, pET32a vector protein (20 μg/mL), and rTgRPP2 at different concentrations (0, 5, 10, 20, 40, and 80 μg/mL) for 48 h at 37 °C, respectively, each well was loaded with the CCK-8 reagents provided by Cell Counting Kit-8 (Beyotime, Shanghai, China). They were cocultured continuously for 2 h, and cell proliferation was then conducted with absorbance using a microplate spectrophotometer at 450 nm (OD450).

### 2.8. Detection of Cell Apoptosis

The Ana-1 cells were diluted to 1 × 10^6^ cells/well in a 12-well plate and preincubated with rTgRPP2 (0, 5, 10, 20, 40, and 80 μg/mL), pET32a vector protein (20 μg/mL), and PBS at 37 °C for 48 h. According to the manufacturer’s instructions, the Apoptosis Detection Kit (Miltenyi-Biotec, Bergisch Gladbach, Germany) was used to check the apoptosis level in murine macrophages with flow cytometry. Before flow cytometry, a macrophage-specific gate was determined by FSC and SSC using the same method mentioned in Section 2.6, and a cross gate was then determined according to the fluorescence minus one (FMO) control.

### 2.9. FITC-dextran Internalization Assay

After being diluted to 5 × 10^6^ cells/well in a 12-well plate, the murine Ana-1 cells were cultured with rTgRPP2 at different concentrations (0, 5, 10, 20, 40, and 80 μg/mL), pET32a vector protein (20 μg/mL), and PBS at 37 °C for 48 h. The cells were washed in cold PBS three times and then collected in PBS and incubated with FITC-dextran (final concentration was 1 mg/mL, molecular weight 40,000 Da; Sigma-Aldrich, Shanghai, China) at 37 °C for 1 h. Then, the murine macrophages were washed three times in PBS, and the FITC-dextran internalization was conducted by flow cytometry to demonstrate the effects of rTgRPP2 on the endocytic ability of Ana-1 cells. After the macrophage subsets were gated by FSC and SSC, median fluorescence intensity (MFI) was calculated.

### 2.10. Nitric oxide (NO) Production Assay

To demonstrate the effects of rTgRPP2 on the secretion ability of murine macrophages, Ana-1 cells were diluted to 5 × 10^6^ cells/well in a 12-well plate and incubated with PBS, pET32a vector protein (20 μg/mL), and rTgRPP2 at different concentrations (0, 5, 10, 20, 40, and 80 μg/mL) for 48 h at 37 °C. Then, the supernatants of the Ana-1 cells were collected, and the Total Nitric Oxide Assay Kit (Beyotime, Shanghai, China), based on the Griess assay [41], was used to analyze the secretion of NO by measuring the absorbance values at 540 nm (OD540).

### 2.11. Identification of the Cytokine Level

According to the manufacturer’s principles, double antibody sandwich ELISA kits (Jinyibai, Nanjing, China) were used to measure the secretion levels of tumor necrosis factor-α (TNF-α), interleukin-1β (IL-1β), transforming growth factor-β1 (TGF-β1), and interleukin-10 (IL-10) produced by the murine macrophages. Referring to the known quantity of recombinant mouse IL-1β, TNF-α, IL-10, and TGF-β1, standard curves were created. The concentrations of cytokines were measured in the supernatant of the Ana-1 cells mentioned in Section 2.9.

### 2.12. Animal Vaccination and Challenge

To investigate the immune protective properties of rTgRPP2, BALB/c mice were assigned to three groups (25 mice per group) at random. Before vaccination, the recombinant protein was dissolved with PBS (the final concentration was 200 μg/mL). Every animal was inoculated intramuscularly with 100 μL of recombinant protein or PBS alone in the leg at different sites, two times, at Days 0 and 14. Two weeks after the last immunization, all animals were challenged with 200 tachyzoites of the *T. gondii* RH strain through intraperitoneal injection. The survival time of each animal was observed and recorded daily.

### 2.13. Determination of the Levels of Antibodies and Cytokines

The animals’ blood was collected from the eye socket at Days 0, 7, and 14 (whole blood was first collected before immunization), and the sera were then separated at once and kept at −20 °C until use.

The levels of antibodies in the sera were investigated by the standard enzyme-linked immunosorbent assay (ELISA) referenced in a previous procedure [42]. In short, the rTgRPP2 protein was firstly dissolved in 50 mM carbonate buffer (pH 9.6) to a final concentration of 100 μL/mL, and each well of the 96-well plate (Costar, Cambridge, MA, USA) was then coated with 100 μL rTgRPP2 overnight at 4 °C. After being washed in PBS containing 0.05% Tween 20 (PBST) three times, the plate was then blocked with PBST containing 3% bovine serum albumin (BSA) for 1 h at 37 °C. The second antibodies HRP-conjugated anti-mouse-IgG, IgG1, and IgG2a (eBioscience, San Diego, CA, USA) were added to investigate the first antibodies in the sera. The immune complex was then incubated with 3,3′,5,5′-tetramethylbenzidine (TMB, Tiangen, Beijing, China). Twenty minutes later, the reaction was stopped by adding 2 M newly prepared H_2_SO_4_. The absorbance was then measured at 450 nm using a microplate photometer (Thermo Scientific, Waltham, MA, USA).

To identify the levels of interferon (IFN) gamma (IFN-γ), IL-4, IL-10, and IL-17 in sera, commercial ELISA kits (Jinyibai, Nanjing, China) were used. Using a microplate photometer, the concentration of cytokines was quantified according to known recombinant mouse cytokines. Each group had five replications, and each serum was quantified once.

### 2.14. Major Histocompatibility Complex (MHC) Molecule Analysis

Mice were euthanized under the supervision of the Animal Ethics Committee, Nanjing Agricultural University, China, and the spleen of each mouse was then collected. A murine lymphocyte separation kit (Solarbio, Beijing, China) was used to obtain the spleen lymphocytes. The obtained cells were randomized into two groups, stained with anti-mouse CD3e-APC (eBioscience, San Diego, CA, USA) and MHC-I-FITC (eBioscience, San Diego, CA, USA) and anti-mouse CD3e-PE (eBioscience, San Diego, USA) and MHC-II-APC (eBioscience, San Diego, CA, USA), respectively, for 30 min at 4 °C. After being washed in PBS three times, the harvested cells were analyzed by a flow cytometer (BD Biosciences, Franklin Lakes, NJ, USA) to determine the levels of MHC class I and II molecules. As described by [40], a lymphocyte subset was gated before analysis, and a cross gate of the lymphocyte subset was then determined according to the FMO controls. Each group had five replications, and each sample was quantified once.

### 2.15. Statistical Analysis

The data are shown as means ± standard deviation (SD). All of the data analysis processes were conducted using GraphPad 6.0 software (GraphPad Prism, San Diego, CA, USA). The deviations among all groups were estimated as significant at *p* < 0.05 using one-way ANOVA followed by Dunnett’s test. The flow cytometry analysis was analyzed using Flowjo software (version 10, Franklin Lakes, NJ, USA). The survival rates were analyzed using SPSS 25.0 software (SPSS Inc., Chicago, IL, USA), and differences in the groups were revealed by the Kaplan–Meier survival test and compared based on the log-rank model.

## 3. Results

### 3.1. Preparation of the Eukaryotic Expression Plasmid

Through double restriction enzyme identification, the recombinant plasmid was successfully constructed, theoretically generating 348 bp and 6229 bp fragments in a 1% agarose gel electrophoretogram (Appendix A). Sequence analysis and alignment with the Blast program showed that the insert in the plasmid was the open reading frame (ORF) of TgRPP2 (Appendix A). All the results confirmed that the plasmid pET32a/TgRPP2 was constructed correctly.

### 3.2. Expression, Purification, and Western Blot Analysis of Recombinant TgRPP2 and the pET32a Vector Protein

The endotoxin levels for purified protein were less than 0.1 EU/mL. The calculated molecular mass of recombinant TgRPP2 was 29.75 kDa, including the pET32a vector protein (18.0 kDa) and the native TgRPP2 protein (11.76 kDa). The sodium salt polyacrylamide gel electrophoresis (SDS-PAGE) results indicated that the molecular weight of the recombinant TgRPP2 protein and the pET32a vector protein was approximately 30 and 18 kDa, respectively (Figure 1a). The recombinant TgRPP2 and pET32a vector protein could be detected by an anti-His-tagged antibody (Figure 1b), and the Western blot analysis showed that both recombinant and native TgRPP2 could be identified by sera isolated from rats challenged with *T. gondii* and sera separated from the rats immunized with rTgRPP2 (Figure 1c,d). Furthermore, the pET32a vector protein could be identified by sera isolated from rats immunized with rTgRPP2 but not sera from rats challenged with *T. gondii* or the sera from normal rats (Figure 1c,d). All the results indicated that recombinant TgRPP2 had high immunogenicity and was involved in triggering an immune response.

### 3.3. Validation of the Binding Capability of Recombinant TgRPP2 with Murine Macrophages

The Ana-1 cells preincubated with rTgRPP2 showed a significant right movement, as can be observed in Figure 2. The percentage of Ana-1 cells in Q2 preincubated with rTgRPP2 was 97.47 ± 0.76%, and it was significantly higher (*p* < 0.001) than the blank (0.66 ± 0.41%) and control (1.87 ± 0.11%) groups. Flow cytometry results indicated that recombinant TgRPP2 could bind on the surface of murine Ana-1 macrophages.

### 3.4. Enhanced Proliferation of Murine Macrophages Triggered by Recombinant TgRPP2

Cell proliferation was assessed and is illustrated in Figure 3. The absorbance values at 450 nm (OD450) revealed that it was highly increased in Ana-1 cells preincubated with 20, 40, and 80 μg/mL of rTgRPP2. No obvious distinctions (*p* > 0.05) in the blank group treated with PBS and the control group treated with pET32a vector protein were observed.

### 3.5. Promoted Apoptosis of Ana-1 Cells Induced by Recombinant TgRPP2

Early-stage apoptosis is the independently ordered cell death controlled by genes in order to maintain cellular homeostasis, while late-stage apoptosis mainly represents necrotic cells. As illustrated in Figure 4, the results indicate that early-stage apoptosis can be significantly induced (*p* < 0.001) by incubation with rTgRPP2 at the levels of 20, 40, and 80 μg/mL. As for late-stage apoptosis, a significant difference (*p* < 0.001) was observed in Ana-1 cells incubated with 10, 20, 40, and 80 μg/mL recombinant TgRPP2 compared to the blank and control groups. Noticeably, late-stage apoptosis could also be induced (*p* < 0.05) by 5 μg/mL rTgRPP2 compared to the blank group. No statistically significant difference (*p* > 0.05) between the blank group and the control group was recorded.

### 3.6. Promoted Phagocytosis in Ana-1 Cells Induced by Recombinant TgRPP2

To determine the endocytic effects on Ana-1 cells, a serial dilution of rTgRPP2 was analyzed on FITC-dextran by flow cytometry (Figure 5). The capacity of endocytic functions of Ana-1 cells was significantly enhanced after culturing with rTgRPP2 at all tested concentrations. In addition, no statistical differences (*p* > 0.05) between the blank group and the control group were detected.

### 3.7. Enhanced NO Secretion of Murine Macrophages Triggered by Recombinant TgRPP2

NO production of murine Ana-1 cells was evaluated and is presented in Figure 6. The Ana-1 cells could drive significantly higher NO production in response to all concentrations of rTgRPP2. The results also demonstrated that no statistical deviations (*p* > 0.05) between the blank group incubated with PBS and the control group incubated with the pET32a vector protein were observed.

### 3.8. Modulation Effects in the Cytokine Secretion of Murine Macrophages

To assess the impact on the cytokine secretion of Ana-1 cells, cytokine levels were measured using the double antibody sandwich ELISA method. When incubated with rTgRPP2, murine macrophages secreted a significantly higher level of proinflammatory cytokines (tumor necrosis factor-α (TNF-α) and interleukin (IL) 1β). As illustrated in Figure 7a,b, the secretions of TNF-α were significantly elevated (*p* < 0.001) at all tested concentrations, and the secretions of IL-1β (Figure 7b) were detected at higher levels at concentrations of 40 and 80 μg/mL. As demonstrated in Figure 7c,d, anti-inflammatory cytokines (transforming growth factor-β1 (TGF-β1) and IL-10) were also detected. The TGF–β1 secretion of murine Ana-1 cells was significantly enhanced when incubated with rTgRPP2 at concentrations of 40 and 80 μg/mL. Furthermore, the IL-10 secretion of murine Ana-1 cells was significantly enhanced (*p* < 0.01) by incubation with 80 μg/mL rTgRPP2. No obvious significance (*p* > 0.05) was observed in the blank and control groups of the four tested cytokines.

### 3.9. Modulation Effects in Antibodies and Cytokine Secretion In Vivo

To investigate the titers of total IgG and isotypes IgG1 and IgG2a, sera harvested from animals were detected using standard ELISA. As demonstrated in Figure 8a, total IgG antibody titers in the rTgRPP2 group were revealed to be remarkably higher (*p* < 0.001) than the blank and control groups at Days 7 and 14. As illustrated in Figure 8b, animals vaccinated with rTgRPP2 showed significantly higher (*p* < 0.01) isotype IgG1 compared to the blank and control groups at Days 7 and 14. As for IgG2a antibody titers (Figure 8c), animals immunized with rTgRPP2 generated a significantly higher level (*p* < 0.01) compared to the blank and control groups at Days 7 and 14. No significant difference (*p* > 0.05) between the blank and control groups was observed; the detailed titers are displayed in Appendix A.

According to the manufacturer’s instructions, the secretions of interferon (IFN) gamma (IFN-γ), IL-4, IL-10, and IL-17 in the sera were investigated based on double antibody sandwich ELISA. As shown in Figure 9a,b, a significantly higher level of IFN-γ and IL-4 was revealed in animals vaccinated with rTgRPP2 at Days 7 and 14. Additionally, animals immunized with rTgRPP2 only exhibited significantly higher (*p* < 0.05) IL-10 secretion at Day 14 compared to the blank and control groups (Figure 9c). As for IL-17, shown in Figure 9d, no remarkable difference (*p* > 0.05) was obtained among the rTgRPP2, blank, and control groups.

### 3.10. The Major Histocompatibility Complex Molecule Changes in Murine Spleen Lymphocytes

Flow cytometry analysis was conducted to reveal the major histocompatibility complex (MHC) molecule changes. As illustrated in Figure 10a, a significant percentage (*p* < 0.001) of MHC class I molecules was revealed in animals immunized with rTgRPP2 compared to the blank and control groups at Days 7 and 14. The MHC class II molecules (Figure 10b) also displayed a significant increase (*p* < 0.01) in animals immunized with rTgRPP2 at Days 7 and 14. No significant difference (*p* > 0.05) between the blank and control groups was observed.

### 3.11. Immune Protection of Vaccinated Mice against Acute T. gondii Infections

To investigate the immune protection of rTgRPP2 against acute toxoplasmosis, mice were challenged artificially with 200 tachyzoites of the *T. gondii* RH strain. As shown in Figure 11, all of the immunized animals succumbed within 16 days. A significant (*p* < 0.05) survival time was revealed in the animals immunized with rTgRPP2 (13.600 ± 0.777 days) compared to the blank (10.400 ± 0.718 days) and control (10.300 ± 0.496 days) groups. Furthermore, no significant difference (*p* > 0.05) between the blank and control groups was observed.

## 4. Discussion

In the present study, the recombinant TgRPP2 protein is expressed by the prokaryotic expression system. Flow cytometry analysis revealed that rTgRPP2 could bind to the surface of murine macrophages and exhibit their effects. After coincubation with rTgRPP2, cell proliferation and apoptosis of murine Ana-1 cells were conducted, and the results indicated that rTgRPP2 could enhance proliferation and expedite apoptosis. The functions of murine macrophages resisting *T. gondii* were subsequently investigated, and endocytic ability and NO secretion were significantly promoted after the samples were cocultured with rTgRPP2. Furthermore, pro- and anti-inflammatory cytokine (TNF-α, IL-1β, TGF-β1, and IL-10) production, exhibited by Ana-1 macrophages, was also investigated, showing the regulation ability of macrophages after incubation with rTgRPP2. The in vivo experiments revealed that rTgRPP2 could trigger immune protection against acute toxoplasmosis. The survival days of the animals immunized with rTgRPP2 were significantly prolonged. Therefore, all these results reveal that recombinant TgRPP2 can modulate murine macrophages in vitro and can provide immune protection against *T. gondii* in vivo.

As a crucial function of macrophages, phagocytosis can indicate the antimicrobial activities of macrophages from another perspective. In the in vitro experiments, proinflammatory cytokines (TNF-α and IL-1β) and NO secretions were significantly promoted. In addition, as a commonly used fluorescent probe to determine the permeability and phagocytic activity of cells, FITC-dextran is recognized by macrophages via various pattern recognition receptors (PRRs) [43]. When the recombinant protein combines with the surface of macrophages, the internalization of dextran appears to be promoted by some PRRs. All the results indicate that the anti-*T. gondii* effects of murine macrophages can be enhanced by rTgRPP2 in vitro. Although M1 macrophages play an important role in suppressing *T. gondii*, the strong proinflammatory response can lead to tissue injury. M2 macrophages are characterized by the production of anti-inflammatory factors and the demonstration of anti-inflammatory activity [44,45]. The anti-inflammatory cytokines (TGF-β1 and IL-10) were only enhanced at a high concentration of rTgRPP2, indicating that a high concentration of recombinant protein can lead to immunosuppression. Therefore, a high-dose vaccination of rTgRPP2 should be avoided.

When the host cells were infected by *T. gondii*, the cell proliferation mechanism of the host cells could be modulated by the parasites [46]. According to previous studies, *T. gondii* invasion can promote the host and neighboring cells from the G1 phase into the S phase [47]. In the present study, the proliferation of Ana-1 macrophages was enhanced by incubation with recombinant TgRPP2 in vitro. These changes were regarded as beneficial for resisting *T. gondii*. However, during *T. gondii* infection, the growth of L6 rat myoblast cells was inhibited, and the proportion of cells remaining in the S and G2/M phases was increased [48]. After the incubation of recombinant histone 4 and murine macrophages for 24 h, cell proliferation was significantly suppressed [49]. Such a difference can be linked to various incubation times, different types of protein, and even the endotoxin level in the purified protein. More research is needed to further clarify the cell proliferation mechanism.

Most protozoan parasite infections can regulate the apoptosis of host cells, subsequently affecting host cell responses [50]. According to previous studies, apoptosis signaling pathways mainly include the NF-κB pathway, mitogen-activated protein kinase (MAPKinase) pathways, c-Jun N-terminal kinases (JNKinase) pathways, and PI3K/PKB/Akt pathways [50,51]. In the present study, recombinant TgRPP2 could induce early-stage apoptosis, offering many survival advantages. Late-stage apoptosis was also enhanced, which may have been due to early-stage apoptosis. Different from necrosis, apoptosis is a rational behavior that sacrifices specific cells for more benefits to the organism, a behavior that is highly controlled by genes [52]. According to previous reports, *T. gondii* has gained the ability to modulate the apoptotic responses of host cells to survive in murine and human cells [53,54]. Furthermore, autoantigens derived from early- and late-stage apoptotic cells can induce the activation of T-cells and the expression of B-cells, resulting in the enhancement of immunity [52,55]. Consequently, the enhanced apoptosis induced by rTgRPP2 could elicit stronger immune protection against *T. gondii*.

The administration dose, route, and strategy play important roles in developing successful immunization [56,57]. In previously published studies, different administration doses, routes, and strategies have been used, but few studies have explained why a definite dose, route, or strategy was chosen. The administration dose has mainly varied from 10 to 200 μg per mouse [58,59,60] in previous studies. In the present study, when incubated with 20 μg/mL recombinant TgRPP2, murine macrophages showed good binding ability with recombinant protein. Furthermore, the murine macrophages exhibited significant proliferation, phagocytosis, NO secretion, and proinflammatory cytokine secretion after incubation with 20 μg/mL rTgRPP2. The cytokines TGF-β1 and IL-10 related to immunosuppression and inflammation remained unchanged under 20 μg/mL rTgRPP2 incubation. Therefore, the administration dose of each mouse was determined to be 20 μg per mouse, leading to final concentrations in peripheral blood of 20 μg/mL (peripheral blood represents approximately 5% of the total body weight of mice). The administration route is a key parameter for triggering effective immune protection by vaccines against *T. gondii* infections [61]. However, a successful vaccine against intracellular parasites should involve a preferable route, so intramuscular administration was selected in the present study. To some degree, the intramuscular route has been preferred as the first choice for investigating immune protection in many studies [62,63,64,65]. Due to a lack of synergy in the vaccine studies of *T. gondii*, the immunization strategy varies in different studies. In the present study, both the primary and secondary immunization triggered significantly higher antibodies (total IgG, IgG1, and IgG2a) and MHC molecules (MHC class I and II). However, such results did not reveal if animals vaccinated with one dose of rTgRPP2 could produce significant immune protection.

In vaccine development, the vaccine targets are of primary importance [66]. *T. gondii* has various life stages in the host and expresses different antigenic determinants during intracellular and extracellular stages, which shows good capacity in immune escape [67]. Faced with numerous antigenic determinants, gene sequence and three-dimensional conformation were mainly used to predict the potential targets [68,69]. However, compared with prediction, in vitro trials are superior. The in vitro results showed rTgRPP2 could regulate the functions of macrophages, indicating the antimicrobial activities of macrophages were promoted in vivo. Hence, in the subsequent trials, the mice were immunized with rTgRPP2 to investigate immune protection in vivo.

The important role of specific antibodies has been proven as the most critical indicator for evaluating adaptive immunogenicity against toxoplasmosis [70,71]. The obtained results indicate that specific antibodies against *T. gondii* can be induced by recombinant TgRPP2, whereas their titers relied on the immunization times. A Th1-biased immune response is essential for effectively resisting the replication of *T. gondii* [72], and the cell- or humoral-mediated immune responses can be evaluated by the ratio of IgG1 and IgG2a antibody titers [73]. In the present study, the antibody titers of IgG1 and IgG2a were obtained from the sera of mice immunized with rTgRPP2, emphasizing that the Th1- and Th2-biased immune responses were activated due to the higher levels of IgG1 and IgG2a. Previous studies have also investigated the types of immune response and obtained variable results. Predominant IgG1 synthesis was evaluated [74] by immunizations with recombinant rhoptry protein (ROP) 5 and ROP18. However, the observations revealed by [60] were different, indicating a predominant Th1-biased immune response when immunizing mice with recombinant ROP5. The possible reasons for this may be related to the different types of adjuvant, administration procedure, and even the immunization dosage.

Activated by MHC molecules, CD4^+^ and CD8^+^ T-cell-mediated cell responses are important for suppressing the replication of intracellular parasite *T. gondii* [75,76]. Antigens presented by MHC class I molecules are mainly endogenous, which can result in strong CD8^+^ T-cell-mediated immune responses [77]. The MHC class II molecules can present more exogenous antigens to CD4^+^ T-cells, which can induce stronger immune responses in resisting *T. gondii* [76]. The flow cytometry analysis in the present study revealed that animals immunized with recombinant TgRPP2 could generate significantly higher levels of MHC class I and II molecules, indicating that endogenous and exogenous antigen presentation was activated. These results also suggest that exogenous recombinant protein (rTgRPP2) could be processed into an antigen peptide and presented to T-cells through the endogenous and exogenous antigen-presenting pathway. In summary, the results reveal that rTgRPP2 is a potential molecular target for vaccines.

As illustrated above, cytokines play an essential role in *T. gondii* resistance via the activation of innate and adaptive immunity [78,79]. Activated innate immunity could further stimulate adaptive immunity, resulting in the clearance of intracellular parasites such as *T. gondii* and, eventually, the development of chronic infection [18]. In the in vivo experiment, the secretion of cytokines IFN-γ and IL-4 was significantly promoted, confirming the Th1- and Th2-biased immune responses. In addition, a previous report reported that the early death of animals infected with acute toxoplasmosis was induced by strong immunity rather than the parasites [80]. Therefore, the secretion of IL-10 cytokines is important. Our studies indicated that IL-10 secretion was significantly enhanced after the second immunization, whereas IL-17 secretion remained constant. These findings indicate that animals vaccinated with 20 μg rTgRPP2 could not yield intense immunosuppression and inflammation and lend credibility to the conclusion that the innate immune system can be activated to oppose the invasion of *T. gondii*.

A direct way to analyze the immune protection of rTgRPP2 is to apply the survival rate of immunized animals against *T. gondii* [81]. In the acute infection model, all challenged mice died of toxoplasmosis, whereas animals immunized with rTgRPP2 revealed significantly longer survival times, demonstrating that rTgRPP2 could provide partial immunity against virulent RH strains of *T. gondii*. The RH strain belonging to *T. gondii* type I (including the GT1 strain) can be isolated from humans and is considered the most virulent lineage in mice [82,83]. As an opportunistic parasite, *T. gondii* can survive in a healthy body and develop into chronic infections, whereas a devastating sequence can be caused in patients with a lower immunity [84]. Further research should evaluate the immune protection of rTgRPP2 in chronic infection individuals.

## 5. Conclusions

Our study is the first to reveal that recombinant TgRPP2 is recognized by murine Ana-1 macrophages and the adherence of rTgRPP2 to Ana-1 cells leads to a series of inflammatory responses that may potentially upregulate the immune response to remove pathogens. The interaction of rTgRPP2 could enhance the immune system to generate antibodies (total IgG and subclass IgG1 and IgG2a), cytokines (IFN-γ, IL-4, and IL-10), and MHC molecules (MHC I and II), along with prolonged survival time. The in vivo experiments indicated that partial immune protection against toxoplasmosis could be provided by immunization with recombinant TgRPP2, indicating that rTgRPP2 could be a candidate for the development of a vaccine and drugs against the replication of *T. gondii*.

## Figures and Tables

**Figure 1 vaccines-09-00357-f001:**
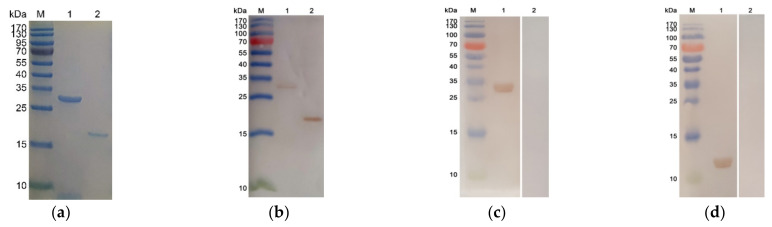
The results of sodium salt polyacrylamide gel electrophoresis (SDS-PAGE) and Western blot analysis. Line M: protein molecular weight marker. (**a**) SDS-PAGE analysis of the purified recombinant *Toxoplasma gondii* ribosomal protein P2 (TgRPP2) and the pET32a vector protein. Line 1: purified recombinant TgRPP2 (rTgRPP2) protein. Line 2: purified pET32a vector protein. (**b**) Western blotting of recombinant TgRPP2 and the pET32a vector protein. rTgRPP2 (Line 1) and pET32a vector protein (Line 2) probed by a mouse anti-His-tagged antibody as the primary antibody. (**c**) Western blot of recombinant TgRPP2. Line M: protein molecular weight marker; Line 1: rTgRPP2 probed by serum from rats against *T. gondii* as the primary antibody; Line 2: rTgRPP2 probed by sera from normal rats as the primary antibody. (**d**) Western blotting of native TgRPP2. Line M: protein molecular weight marker. Line 1: total soluble protein of *T. gondii* tachyzoites probed by sera from rats immunized by rTgRPP2 as the primary antibody. Line 2: total soluble protein of *T. gondii* tachyzoites probed by sera from normal rats as the primary antibody.

**Figure 2 vaccines-09-00357-f002:**
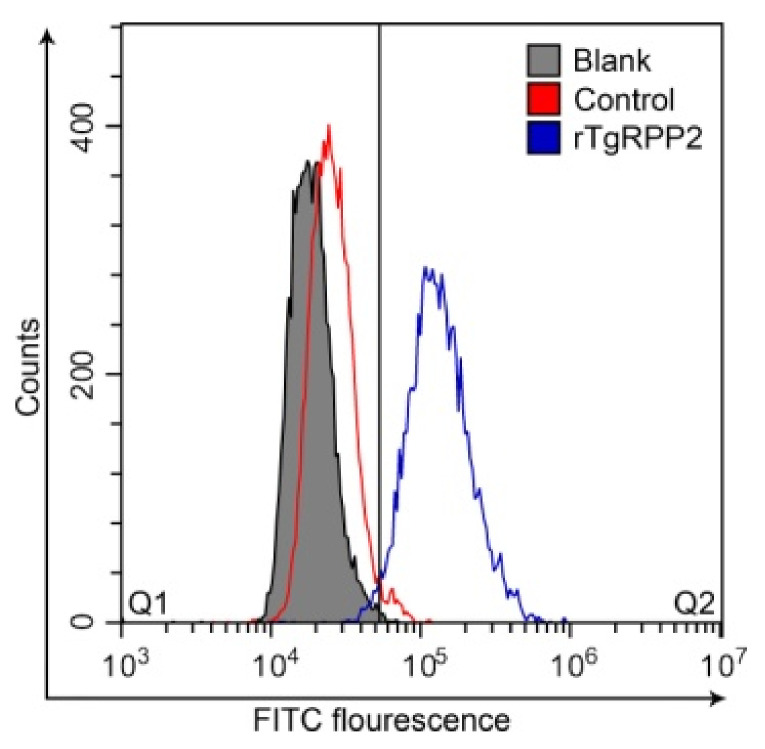
Flow cytometric analysis of rTgRPP2 binding to murine Ana-1 macrophages. In the blank and control groups, the Ana-1 cells were treated with phosphate-buffered solution (PBS) and pET32a vector protein (20 μg/mL), while the experimental group was treated with purified recombinant TgRPP2 (20 μg/mL; *n* = 3).

**Figure 3 vaccines-09-00357-f003:**
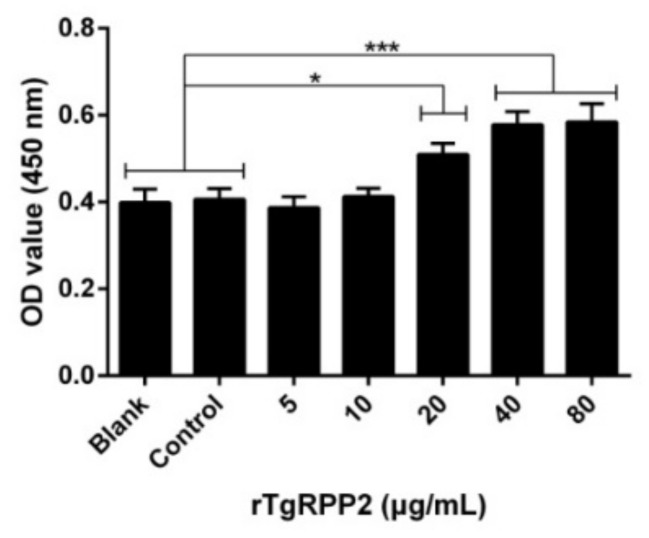
Effects of the proliferation of different concentrations of rTgRPP2 on murine macrophages. The cell proliferation index was calculated by the absorbance values at 450 nm (OD450) values. The values were evaluated using one-way ANOVA analysis, followed by Dunnett’s test, and presented as the mean ± standard deviation of three independent experiments (*n* = 3). * *p* < 0.05 and *** *p* < 0.001 compared with the blank group or the control group.

**Figure 4 vaccines-09-00357-f004:**
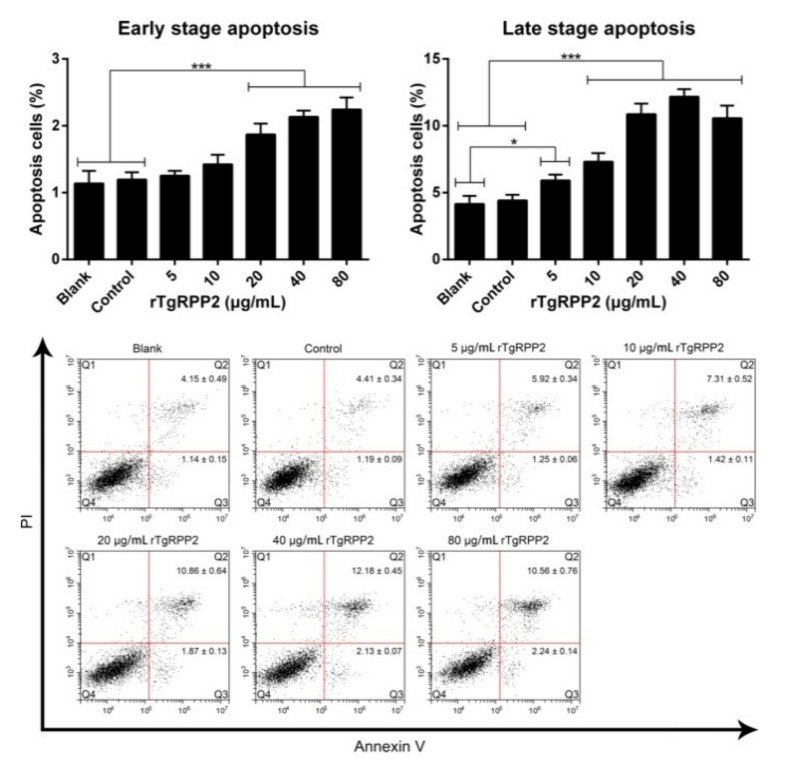
Recombinant TgRPP2 induces the apoptosis of murine Ana-1 cells. Ana-1 cells were preincubated with different concentrations of rTgRPP2 for 48 h. Quadrant Q3 represents early-stage apoptosis (annexin V+/PI−), while quadrant Q2 represents late-stage apoptosis (annexin V+/PI+). Values were evaluated using one-way ANOVA analysis, followed by Dunnett’s test, and expressed as the mean ± standard deviation of three independent experiments (*n* = 3). * *p* < 0.05 and *** *p* < 0.001 compared with the blank group or the control group.

**Figure 5 vaccines-09-00357-f005:**
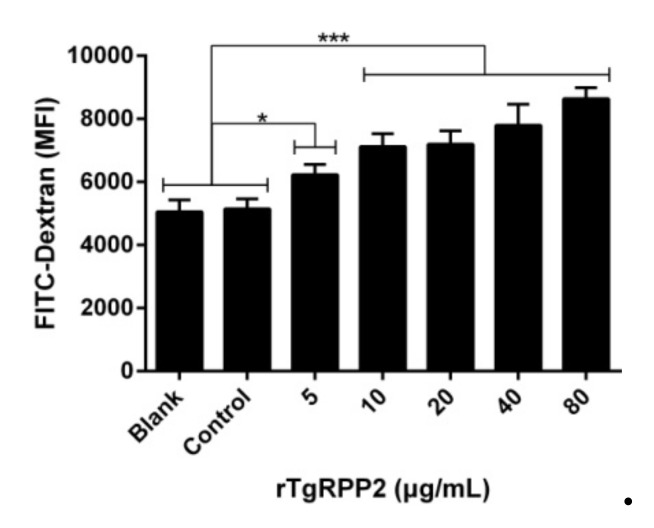
Recombinant TgRPP2 significantly affects the phagocytosis of murine macrophages. Median fluorescence intensity (MFI) was established based on the statistical data. Values were evaluated using one-way ANOVA analysis, followed by Dunnett’s test, and presented as the mean ± standard deviation of three independent experiments (*n* = 3). * *p* < 0.05 and *** *p* < 0.001 compared with the blank group or the control group.

**Figure 6 vaccines-09-00357-f006:**
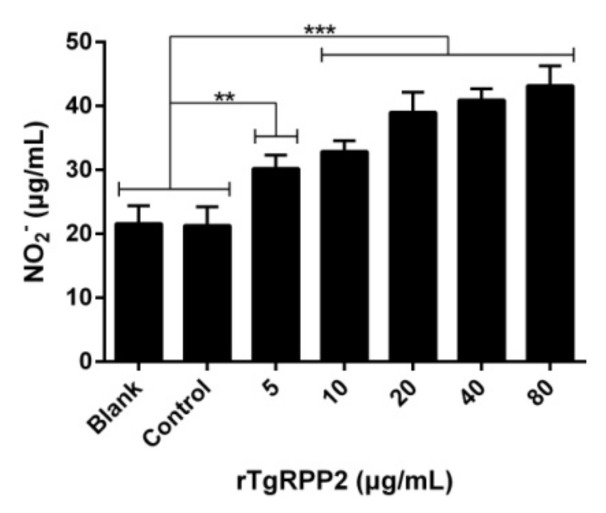
Effect of rTgRPP2 on the NO production of murine Ana-1 cells. Through treatment with different concentrations of rTgRPP2 for 48 h, the NO concentrations were determined. Values were evaluated using one-way ANOVA analysis, followed by Dunnett’s test, and presented as the mean ± standard deviation of three independent experiments (*n* = 3). ** *p* < 0.01 and *** *p* < 0.001 compared with the blank group or the control group.

**Figure 7 vaccines-09-00357-f007:**
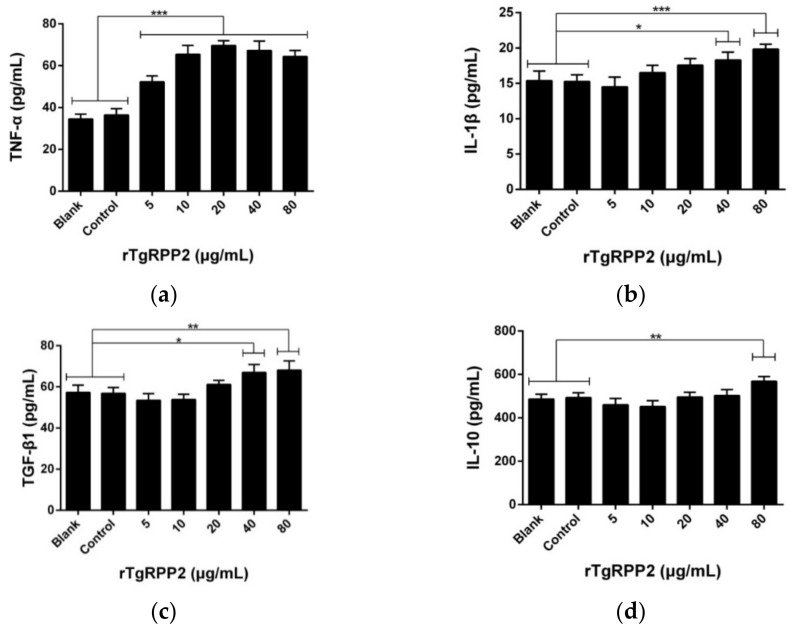
Effect of recombinant TgRPP2 on tumor necrosis factor-α (TNF-α) (**a**), interleukin-1β (IL-1β) (**b**), transforming growth factor-β1 (TGF-β1) (**c**), and interleukin-10 (IL-10) (**d**) secretion of murine Ana-1 cells. The Ana-1 cells were preincubated with rTgRPP2 at different concentrations for 48 h, and the cytokine secretion was then measured by ELISA kits. Values were evaluated using one-way ANOVA analysis, followed by Dunnett’s test, and presented as the mean ± standard deviation of three independent experiments (*n* = 3). * *p* < 0.05, ** *p* < 0.01, and *** *p* < 0.001 compared with the blank group or the control group.

**Figure 8 vaccines-09-00357-f008:**
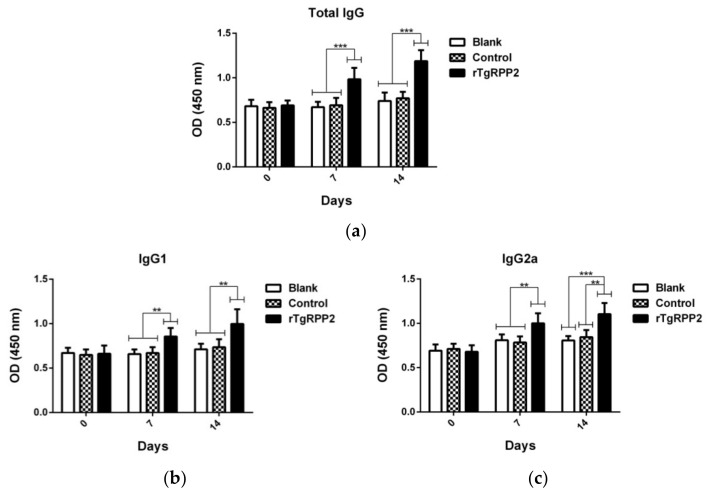
Determination of total IgG (**a**) and subclass IgG1 (**b**) and IgG2a (**c**) antibodies in the sera from animals vaccinated with PBS, pET32a vector protein, and rTgRPP2. Results were evaluated using one-way ANOVA analysis, followed by Dunnett’s test, and shown as the mean of OD450 ± standard deviation (*n* = 5). ** *p* < 0.01 and *** *p* < 0.001 compared with the blank group or the control group.

**Figure 9 vaccines-09-00357-f009:**
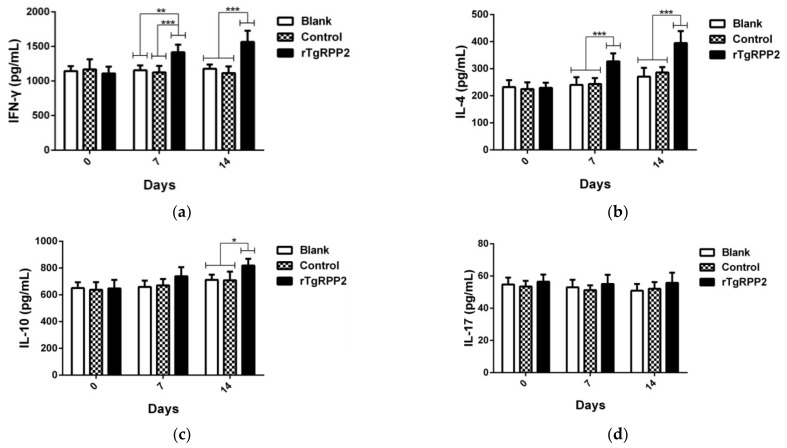
Cytokine production. Double antibody sandwich ELISA kits were used to determine the level of IFN-γ (**a**), IL-4 (**b**), IL-10 (**c**), and IL-17 (**d**) in sera from animals. Results were evaluated using one-way ANOVA analysis, followed by Dunnett’s test, and values are shown as the mean ± standard deviation (*n* = 5). * *p* < 0.05, ** *p* < 0.01, and *** *p* < 0.001 compared with the blank group or the control group.

**Figure 10 vaccines-09-00357-f010:**
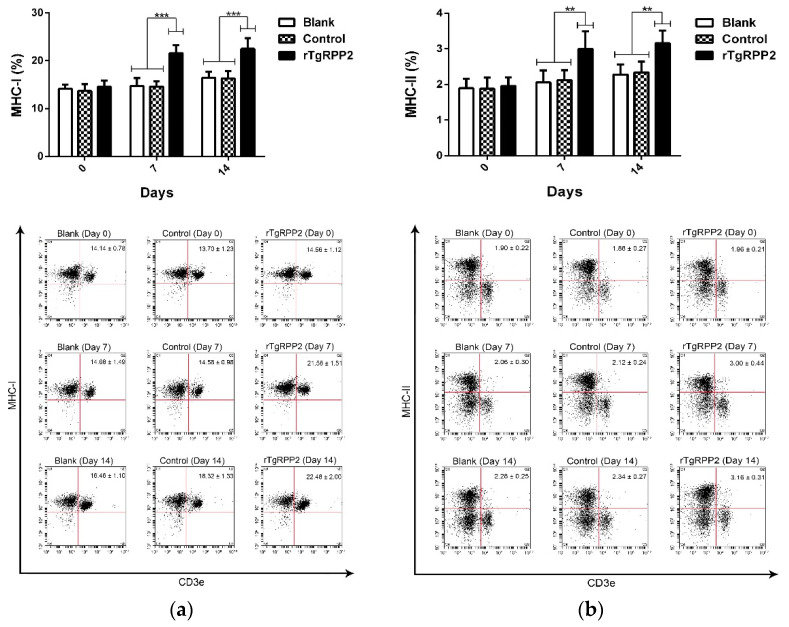
Flow cytometry analysis of major histocompatibility complex (MHC) class I (**a**) and MHC class II (**b**) molecules in murine spleen lymphocytes. Results were evaluated using one-way ANOVA analysis, followed by Dunnett’s test, and values are shown as the mean ± standard deviation (*n* = 5). ** *p* < 0.01 and *** *p* < 0.001 compared with the blank group or the control group.

**Figure 11 vaccines-09-00357-f011:**
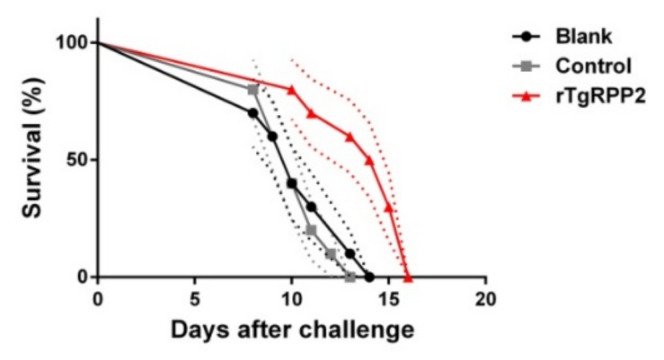
Mice survival rates after challenge infection with the *T. gondii* RH strain. Results were analyzed by the Kaplan–Meier test and compared based on the log-rank model. Values are shown as the mean ± standard deviation (*n* = 10). The dotted lines represent standard deviation.

## Data Availability

The data presented in this study are available within the article and Appendix A.

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
