# Peer review of "Recombinant Toxoplasma gondii Ribosomal Protein P2 Modulates the Functions of Murine Macrophages In Vitro and Provides Immunity against Acute Toxoplasmosis In Vivo"

_vaccines, 2021, doi:10.3390/vaccines9040357_

Round 1
Reviewer 1 Report
The paper entitled “Recombinant Toxoplasma gondii ribosomal protein P2 modulates the functions of murine macrophages in vitro and provides immunity against acute toxoplasmosis in vivo” by ZhengQing Yu and colleagues describes the expression and purification of the T. gondii ribosomal protein P2 (TgRPP2) and its immunological effect in vitro (murine Ana-1 cells) and in vivo (Sprague Dawley rats). T. gondii infection is a major problem in veterinary and human medicine and new therapeutic and prophylactic tools are needed. This paper contributes to this propose.
Major comments
The choice of the immunological assays used to assess the response to rTgRPP2 stimulus is not rationally justified. I recommend including in the introduction an explanation about this topic, describing some mechanisms used by the immune system to control the parasite. This initial explanation will support the choice of the immune assays used in this work. Some of the information included in the discussing section would be transferred to introduction.
Lines 27 and 28- Abstract- “All of the results indicated that murine macrophages could be regulated by rTgRPP2 and are essential for the elimination of T. gondii and the maintenance of tissue homeostasis.” and Lines 83-85- Introduction “The in vivo results revealed that rTgRPP2 could provide immune protection against the replication of T. gondii.” I do not think that the assays carried out allow us to conclude that rTgRPP2 is essential for the elimination of T. gondii and to prevent protozoa replication. None of the assays evaluated the parasitic load or parasitic viability, so the conclusions are not supported by the results. I recommend greater restraint in the study's conclusions.
Minor comments
Line 43- The sentence “As a T. gondii type I strain [14], the S48 strain is virulent in mice.” seems to me out of context.
Line 68- falciparum was highly homologous, remove italic, please
Line 97- I suggest to replace “consisting of” by supplemented with
Line 147- “1: 5000”- remove space, please
Line 187- “μg/mL)m” m????
Lines 191-194- “After having been bathed in PBS three times to remove the uncombined antibodies, the murine macrophages were then analyzed by flow cytometry (Beckman Coulter, California, USA). The forward scatter (FSC) and side scatter (SSC) were taken to gate the lymphocyte subsets [32].” If I understood correctly, the assay was performed with a pure macrophage culture (Ana-1 cells). Don't you mean macrophage subset (or leucocyte subsets) instead of lymphocyte subsets? The same problem in line 211 and 222!
Lines 156-161- “In total, 200 μg of purified rTgRPP2 was emulsified with an equal volume of Freund’s complete adjuvant (Sigma-Aldrich, Saint Louis, USA) to produce an emulsion vaccine, and the SD rats were then immunized with the emulsion vaccine. Applying two-week intervals, the emulsion mixture of 200 μg of purified rTgRPP2 and an equal volume of Freund’s incomplete adjuvant (Sigma-Aldrich, Saint Louis, USA) was injected into rats four times.”- It seems that the idea is repeated!
Lines 243-246- Before vaccination, the recombinant protein was dissolved with PBS (the final concentration was 200 µg/mL). Every animal was inoculated intramuscularly with 100 µL of recombinant protein, or PBS alone at different sites, two times at day 0 and 14.”
The two explanations (Lines 156-161 and Lines 243-246) are not coincident. Where (in the rat body) was the vaccine administered?
Lines 347-353- The concepts of early stage and late stage apoptosis must be explained in the text and its impact in the immune response discussed (in the discussion). The authors mention in Lines 521-522 “Consequently, the enhanced apoptosis induced by rTgRPP2 could promote the elimination of T. gondii.”, but I question whether the induction of apoptosis by T. gondii cannot be a mechanism of parasite evasion to the host's immune system, insofar as the apoptotic infected cells can be phagocytized by other naive cells.
Good look
Best regards

Reviewer 2 Report
The submitted manuscript, Yu et al, is an interesting manuscript about immunological study of TgRPP2 protein. In this manuscript, the authors expressed the protein in BL-21 cells and confirmed binding of recombinant TgRPP2 with Ana-1 cells using flow cytometry. Cell proliferation analysis and apoptosis study confirmed that TgRPP2 interaction with Ana-1 indeed leads to series of inflammatory responses. The manuscript is well written and describe all sections in explicit details. This paper represents an important contribution to Toxoplasma community and will be useful for the broad audience of MDPI vaccines.
Minor comments:
Figures and Tables:
- Figure 1a and b: In panel a, TgRPP2 band is much intense than pET32a vector protein. But in panel b (western blot using anti-His antibody), TgRPP2 band is relatively faded as compared to the vector protein. Please comment.
- Figure 11: Please explain dotted lines in the graph.
Reviewer 3 Report
I think the work is really interesting and innovative.
I congratulate you on the overall approach of the paper and on how well it is done in the chapter 3.3 "Validation of the binding capability of recombinant TgRPP2 with murine macrophages" .in pages 8 -12, as well as I am pleasantly surprised to have entered the Supplementary Materials.
Also very interesting as it is scientifically very useful presented in the Chapter 4, Discussion .
Given the great validity of the work presented , I ask you , please, to add only a few additional considerations in the conclusions (Chapter 5).
Reviewer 4 Report
The present study reports the potential protective effects of a T.gondii protein to be considered as vaccine candidate. Immune response characterization in vivo and in vitro against this protein has been developed in animal models.
The manuscript has been clearly improved but there are still some technical issues that need to be sorted:
-BCA method is not mencioned above, as inducated. This methodology needs to be included.
- To correct: Luria Bertini, Luria Bertani
- In animal and cell culture description, media composition is missing, only FBS and antibiotics mentioned.
- Imnunization to obtain the serum to perform some og the assays, either is missing or it is in the incorrect place. Authors explain western blot and after that they explain inmunization and ELISA for serum titration. This must be explaied before in order to have a clear and well organised methodology.
Ethics committee aproval is missing. Animal experiments must be carried out following regulatory permissions.
Finally, the discussion part is very long and with too many references to bibliography to disscuss minor related topics. This must be revisited to build a strong evidences based discussion, from the results obtained and make the text more clear, precise and summarised. Ñ
Round 2
Reviewer 1 Report
Thank you! I appreciate the clarifications. However, I still have some doubts.
In the sentence “According to the instructions, the early-stage apoptosis means the normal physiological process routinely carried out in organisms, while the late-stage apoptosis mainly represents the necrotic cells.”, please remove “According to the instructions” and improve the definition of early stage apoptosis. In the caption of Fig 4, please include the phenotype of early apoptotic cells (Annexin + Pi-) and late apoptotic cells (Annexin + Pi+).
The text: "Unlike cell death, apoptosis a active and rational behavior to sacrifice specific cells for more benefits of the organism [57]. Furthermore, cell apoptosis does not lead to membrane integrity loss and an inflammatory reaction, and this is an essential physiological process that plays an important role in tissue homeostasis during T. gondii infections [50]. These autoantigens released from early-stage apoptotic cells can induce the activation of T cell and expression of B cell, finally generates autoimmunity [57,58]. Different from the apoptotic cells, the necrotic cells can induce partial cells to proliferate and differentiate into giant phagocytes [59]. Consequently, the enhanced apoptosis induced by rTgRPP2 could elicit stronger immune protection and provide protection against T. gondii." is not clear enough! The objective is to explain the benefit of apoptosis.
I think the text needs a review of the English language.
Best regards
Author Response
Please see the attachment.

This manuscript is a resubmission of an earlier submission. The following is a list of the peer review reports and author responses from that submission.
Round 1
Reviewer 1 Report
This paper is focused on resolving the immunomodulatory properties of Toxoplasma ribosomal protein P2 and whether it provides immunity against acute toxoplasmosis in mice. This work follows from previous studies on RPP2 from both Plasmodium falciparum and Toxoplasma gondii. PfRPP2 traffics to RBC membrane and antibodies against it block Pf invasion. While TgRPP2 does not traffic to the membrane of HFF host cells, it is detected on the surface of tachyzoites and antibodies against it block invasion. Prior to this work, it was unclear if TgRPPR2 had immunomodulatory activity.
To assess the potential of TgRPP2 to initiate an immune response, investigators purified recombinant TbRPP2 from E. coli and measured its effect on several macrophage activities. Specifically, they tested whether the protein could (i) bind to murine macrophages, (ii) enhance proliferation, (iii) promote apoptosis and phagocytosis, (iv) enhance NO production and alter antibody levels and cytokine expression in vitro. They also follow on with in vitro experiments to measure immune responses of mice primed with rTgRPP2. IgG antibodies, IFN-gamma, IL-4, IL10, MHCI and MHCII were increased in mice treated with rTgRPP2 while IL17 remained unchanged. While treatment with rTbRPP2 did not protect mice from infection it increased the time to survival.
While the experiments are logical, it was hard at times to assess the science as the grammar was extremely hard to navigate. I strongly suggest authors consult with an editor prior to resubmission. The errors were too numerous for me to track and many times I was unclear as to whether I understood what the authors were trying to articulate. Fortunately, the experiments were straightforward and based on commercially available kits so I was able to follow most of them. In figure 2 and on line 588 it is unclear what the “primary antibodies” are. Did they use serum from infected rodents or antibody generated against RPP2? While many of the responses are statistically significant, it is unclear whether they are biologically relevant. Many of the effects are mild and it would be helpful to have a positive control to determine how the TgRPP2 response compares to one that is well characterized and known to initiate a robust immune activation against Toxoplasma. Perhaps it is enough for the authors to comment on how the the effect of RPP2 is compared to other molecules that initiate a strong response.
Reviewer 2 Report
In the manuscript entitled “Toxoplasma gondii ribosomal protein P2 modulates the functions of murine macrophages in vitro and and provides immunity against acute toxoplasmosis in vivo” the authors characterized the immunomodulatory effects of T. gondii ribosomal protein P2 (TgRPP2). Below are the suggestions to improve the manuscript.
- Line 23: The results indicated that rTgRPP2 could bind to murine Ana-1 cells and showed a good reactionogenicity.
- Line 38: As an important zoonotic protozoan, T. gondii, which causes toxoplasmosis, has infected more than 1 billion people around the world at a conservative estimate [2,3].
- Lines 39 & 40: As a smart opportunistic parasite, tachyzoites of T. gondii can develop into bradyzoites when its transmission is inhibited [4].
- Lines 41 & 42: Normally, immunocompetent patients do not display symptoms, however these parasites could cause severe disease even leading to death in immunocompromised individuals (5).
- Lines 43 & 44: According to the previous reports, T. gondii could spread by ingestion of unsterilized meats, contaminated fruits, vegetables, even drinking water [6-8].
- Lines 45 & 46: Moreover, vertical transmission of T. gondii could cause irreversible damage to the fetus, and abortion, stillbirth and congenital deformities might occur [9-12].
- Lines 46-52: Currently, there are no effective vaccines and treatments for T. gondii, only Pyrimethamine (PYR) and sulfadiazine (SDZ) are approved for the treatment of T.gondii, but these two drugs can only depress Toxoplasma folate synthesis [13], and have no effect on bradyzoites suggesting that these drugs cannot eliminate chronic infection [14]. Besides, the side effects such as kidney disorders, immunosuppression, teratogenicity have been reported [15,16].
- Lines 52 & 53: Clearly, it is urgent and important to develop effective prevention and treatment strategies against T. gondii.
- Lines 62-65: A recent study had shown that the ribosomal P2 protein and P protein pentamer complex were immunogenic in Plasmodium falciparum [23], and P2 protein from P. falciparum is highly homologous to T. gondii RPP2 protein (TgRPP2).
- Lines 65 to 69: Unlike ribosomal P2 protein of P. falciparum [24,25], the TgRPP2 protein has been demonstrated to exist on the surface of T. gondii tachyzoites by using immunohistochemical method, however, the TgRPP2 protein was not observed on the surface of HFF cells during T. gondii infections in vitro [22]. Hence, this finding strengthens the hypothesis that TgRPP2 might be involved in T. gondii invasion.
- Lines 71-73: Although the signal and transmembrane domains were not reported in the TgRPP2 protein [24], there are still some reports suggesting that TgRPP2 might be involved as an invasion ligand, which could help T. gondii adhere to host cells [26].
- Lines 74-77: Using prokaryotic expression system to further explore how recombinant TgRPP2 (rTgRPP2) modulates host immunity, we employed an unbiased approach to investigate the potential effects of rTgRPP2 on murine macrophages in vitro and the immune protection against acute toxoplasmosis in vivo.
- To investigate the effects of rTgRPP2 on macrophage apoptosis, Ana-1 cells were incubated with various concentrations of recombinant protein (0, 5, 10, 20, 40, 80 μg/ml) at 37°C for 48 h. Then according to the instructions of the Annexin V-FITC kit, flow cytometry was employed to determine the effect of rTgRPP2 on apoptosis (Figure 4).
- Figure 4. Recombinant TgRPP2 induces the apoptosis of murine Ana-1 cells. Ana-1 cells were preincubated with different concentrations (0, 5, 10, 20, 40, 80 μg/ml) of rTgRPP2 for 48 h. Using an Annexin V-FITC kit, apoptosis was measured by flow cytometry. The Ana-1 cells in the blank groups were treated with PBS while the control group cells were treated with pET32a vector protein. Results were evaluated using one-way ANOVA analysis followed by Dunnett's test and expressed as mean ± standard deviation of three independent experiments. *P < 0.05 and ***P < 0.001 compared with blank or control group.
- The authors should depict the results as both percentages and MFI.
- The rationale for the assessment of MHC-I and MHC-II molecules is not clear. The authors should have characterized CD4 and CD8 T-cell frequencies instead.
- Whether proper isotype controls were used in the flow cytometry experiments? What markers were used to identify mouse macrophages? What gating strategies were used? The authors should describe these points in detail and the gating strategies used should be depicted in the corresponding figures.
- The authors should have validated in vitro findings using peritoneal macrophages from acute toxoplasmosis experiment ex vivo.
- For the survivability experiment, the authors should perform atleast two independent experiments.
Reviewer 3 Report
The authors generate a recombinat of a ribosomal protein of Toxoplasma gondii and study its involvement in immunity in vitro and in vivo. Although the paper is of interest to the field and the authors made a great effort developing several experiments, there are several things that must be improved in order to accept the study. As it is a work in which they developed several immunity related studies and researched immune response at a quite deep level, the introduction must include some immunology related information to provide the reader enough information to follow the study. Regarding material and methods, the experimental design related to in vivo assays lacks a lot of information as well as the ethical part of animals used for the experiments. This must be clearly improved. About the in vivo challenge, the authors seem quite positive about the result because they delay the death of the animals for about 3 days. This may be an indicative of immune response involvement, as suggested by other experiments but this assay does not necesarily involve that. Such conclusion cannot be extracted directly from that assay. Moreover, authors should also include error bars in the graph and not only the data in the figure legend.Author Response
Please see the attachment.

Reviewer 4 Report
The authors address in this manuscript the issue of isolation and characterization of a ribosomal protein usable in creating an effective vaccine against T. gondi. It is a very topical issue; the manuscript is interesting, the research is well done, the results are encouraging, but some adjustments of the paper are needed.
lines 36-39. I suggest the rephrasing, to be more concise and with shorter, more understandable sentences. E.g.: Toxoplasma gondii (T. gondii) is an obligate intracellular parasite belonging to the phylum Apicomplexa. It can infect a wide range of warm-blooded animals, posing a zoonotic potential and being responsible for human toxoplasmosis. According to a conservative estimation, more than 1 billion people had been infected around the world.
lines 39-40. "As a smart opportunistic parasite...". Well, opportunistic parasites are usually symbiotic organisms which become pathogenic in specific conditions, mainly in immunocompromised hosts (e.g. Balantidium coli). T. gondii isn't an opportunistic pathogen; it is a true pathogen, that can cause itself a disease, without favourable conditions. Please, rephrase!
lines 41-43. Please, more attention in phrasing! More correct: "Normally, no symptoms develop in immunocompetent patients, but the parasites can cause severe disease, even death, in immunocompromised individuals.".
lines 43-44. Better: "According to the previous reports, T. gondii can spread by raw meat, fruits, vegetables, even drinking water.".
lines 45-46. I propose: "Moreover, vertical transmission of T. gondii can cause irreversible damage to the fetus, abortion, stillbirth; additionally, congenital deformities can occur.". It sounds better!
lines 46-47. "Nowadays, there were no effective vaccines...". Why did you associate the present time (nowadays) with a past tense of the following verb (there were no)?
lines 46-52. Well, an extremely confusing and difficult to understand phrase. I propose: "Nowadays, there are no effective vaccines and treatments for T. gondii. However, Pyrimethamine (PYR) and sulfadiazine (SDZ) were approved for the treatment of toxoplasmosis. Still, these two drugs only depress Toxoplasma folate synthesis and do not affect the bradyzoites, suggesting that it could not eliminate the chronic infection. Besides, the side effects such as kidney disorders and immune inhibitions, teratogenicity on the fetus were also obvious.". Sounds incomparably better!
lines 52-53. I propose: "Conclusively, the development of an effective prevention and treatment therapy against T. gondii is an urgent and important requirement!". It sound better!
line 54. "As we all know...". Better: "As it is known...". More, on the same line: "...is composed..." instead of "was composed". A complete ribosome still contains the same subunits!
After reviewing the first paragraph of the Introduction, I understand that your article must be rewritten. Particular emphasis must be placed on English grammar! Sorry, but I am not usually the one that handles rewriting your manuscript.
Moreover, reading the whole article, I realized that there is a big disarray in it. Elements related to Material and Methods are exposed in the Results section. The Results section is extended to 8 pages and includes unnecessary items. Practically, each subchapter of Results starts with the description of the methods and materials used. However, there is a section specifically designed to describe these methods. The Materials and Methods sections, respectively Results, must be completely reorganized.
Reviewer 5 Report
The authors identify and express a novel T. gondii ribosomal protein, TgRRP2, and examine the proteins effect as an immunomodulator by exposing murine macrophages to the recombinant protein. They determine that this protein may have a role as a vaccine candidate based on preliminary in vivo experiments. The work is interesting, and the choice of RRP2 appears well supported and the statistics well performed throughout. However, although the work shows promise, there are some major issues which should be addressed before publication.
Please define exactly what the 'pET32a vector protein' is, this is a key control but there is no information about what the protein is. Is this a prokaryotic protein that macrophages could recognize?
Line 47 – There is a vaccine against T. gondii (Toxovax, Buxton et al 1991, Vet Rec.) that is licensed for use in animals, please introduce this here.
Figure 1 b and c – These western blots do not contain the correct controls. In Fig 1b, the ‘pET32a vector protein’ should be ran on the same gel as the rTgRPP2 and probed with the sera of the animals immunized with RPP2. This may react, since it appears to be the tagging sequence, but should run at a different size which would allow you to distinguish. Also, as the vector likely includes a tag, it would be good to include a western blot using an antibody against the tag (ran on both proteins) to confirm that you are looking at the right thing. For Fig. 1d, western blots with two different antibodies should not be combined like this, and it would be important to have a negative control here, e.g. the same amount of protein from human cells, to confirm that the antibody is specific.
Figure 2. As mentioned below, this experiment should be repeated at least three times. It would be more useful to see a dose response as well, e.g. incubating with 5, 20 and 80 ug/ml to ensure that the response is real.
Figure 3. How much protein is in the control sample? It is possible that this protein is having effects on the cells, and if it is only 20ug/ml then this may affect your results. These experiments should be repeated with the same amount of control as the highest test sample (e.g. 80 ug/ml). There is also no reason to compare with the blank group, better to compare to the control.
Figure 4. I have the same point as above with regards to the control. I am also confused on how incubation with TgRRP2 causes both apoptosis and cell proliferation, do the authors have a potential explanation on how this could occur?
Figure 5. FITC-Dextran is not a good way to measure phagocytosis as it is mostly taken up through endocytosis/pinocytosis. Other assays such as FITC-labelled heat killed E. coli or zymogen would be more appropriate here. Also, what would be the mechanism for how RRP2 affects macrophage phagocytosis rates?
Figure 6. See my criticisms for figure 3 and 4, however for this assay, were values normalized to the number of cells (or total amount of protein lysate)? Figure 3 shows that addition of RRP2 causes increased cell division so you would expect more cells in the treated samples, which could affect the results. This should be normalized to the cells, potentially by taking another well on the plate, treating with the same concentrations of RRP2, lysing and measuring protein concentration using BCA assay.
Figure 7. See figure 6, the number of cells could also affect these results. Did you measure any cytokines which were not affected by RRP2 incubation?
Figure 7 and Figure 9. Why where different cytokines measured in each experiment? Why did you choose these specific cytokines to measure and why are they important? What do these changes mean for the immune status of the animal and how do they fit in the broader context of Toxoplasma infection?
Discussion – The discussion requires more work, the questions that should be addressed are, what previous work in this area has been done? For example, have other T. gondii proteins been assessed for their ability to act as vaccine candidates? How does RRP2 compare with these? Are the responses seen from the assays chosen what would be expected? Are any unexpected? What are the features of RRP2 that would make it a good candidate?
Minor issues
General points
In the results, you don’t need to list all of the values which are displayed in the tables, it makes the manuscript more difficult to read and when you measure OD or fluorescent intensity the numbers are not comparable in any case. Best to remove most of them unless you which to draw attention to specific changes in the text
Why were all experiments done at 48h post addition of the protein? Did you test other time points? If so, this would be valuable, 48h is quite a long time for these kind of assays and it would be interesting if responses were seen more rapidly.
Figure 1a legend – you don’t need to specify which ladder you used here, or how the protein was purified, this information is better in the materials and methods.
Line 54 – Although you are probably correct, please remove ‘ As we all know,’
Line 65 – Is TgRRP2 the same as R2 which was previously published in Sudarsan et al 2015? If so, what is the reason for changing the name? Also, please include the toxoDB ID.
Section 2.1 – This should mostly be moved to the material and methods section, and figure 1A should be combined with figure S1
Section 2.3 – There is no reason to state the values obtained here, FACS measures in arbitrary units so these numbers are not meaningful. However, you should specify how long the cells were incubated with the protein, and how many times the experiment was repeated. This should be repeated at least three times, however you can show representative results from a single experiment.
Line 128 – This should be rewritten to: Cell proliferation was assessed using the CCK-8 kit
Figure 8. Please indicate how many animals these results represent
Line 368-377 – This section is not required, please remove it.
Line 579 – The molecular weight of dextran should be given, as well as the manufacturer and ideally catalog number.
Round 2
Reviewer 1 Report
The authors have addressed my concerns
Author Response
Thank you very much for your great efforts on our manuscript, and we will try our best to improve the manuscript continuously.
ZhengQing Yu
E-mail: 2018207044@njau.edu.cn
Corresponding author: XiangRui Li
E-mail: lixiangrui@njau.edu.cn
Reviewer 2 Report
The authors have addressed the concerns satisfactorily.
Author Response

(The authors gave the same response as above.)

Reviewer 3 Report
After revisions, authors have clearly improved the quality of the manuscript. They have included most of the requested and introduced significative changes in the paper, being now better organised and more clear to the reader.
Material and methods as well as introduction and results are described in more detail in the new version.
Despite all the changes I suggest they re-write a more precise conclusion about the data regarding in vivo analysis, in a way that evidences their hyphotesis. Their hyphotesis about immune protection is not completely supported by the data. They need to base on evidences and extract a more precise conclusion.
Reviewer 4 Report
You have considered all my criticisms.
Author Response

(The authors gave the same response as above.)

Reviewer 5 Report
Only one of my material critisisms have been addressed (the western blot controls). However, the authors attempts to wave away the rest of my points are not convincing. For example, no rational was given for why endocytosis (not phagocytosis) was measured, of what effect a possible change in rate of endocytosis would mean for the immune response to this project. They also refer to it as measuring phagocytosis still in the discussion, which is not true.
The authors mention that more research is needed into the times and concentrations of protein incubation and I agree, however some of it should be done in this manuscript.
Further, although they show that incubation with their protein changes cell proliferation (and, confusingly, apoptosis) they do not take this into account in any of their further assays. Saying that this represents the in vivo situation is clearly not correct.
Beyond this, many of the assays the changes are negiliable and unlikley to be relelvant in vivo. For example, IL-1B is increased a small amount (perhaps 25%) but incubation with heat killed parasites leads to more than a 100x increase.
Round 3
Reviewer 5 Report
The authors performed no additional experiments. I do not this this work should be published in its current form.